# Effect of Banking Time Intervention on Child–Teacher Relationships and Problem Behaviors in China: A Multiple Baseline Design

**DOI:** 10.3390/bs14030213

**Published:** 2024-03-06

**Authors:** Zedong Zhang, Ye Wang

**Affiliations:** 1Faculty of Education, Northeast Normal University, Changchun 130024, China; 2School of Preschool Education, Changchun Normal College, Changchun 130216, China; alucie@foxmail.com

**Keywords:** problem behaviors, child–teacher relationships, banking time intervention

## Abstract

A positive child–teacher relationship is a crucial means of addressing problem behaviors in young children. In recent years, there has been an increase in factors triggering problem behaviors in young children. It is particularly important to employ universally applicable and scientifically effective strategies to improve child behavior. Banking Time, as an emerging variant of play therapy, aims to enhance child behavior by establishing an intimate child–teacher relationship. This study conducted a multiple-baseline experiment involving eight four-year-old children and their teachers from China, exploring the effectiveness of Banking Time in enhancing child–teacher relationships and subsequently improving child behavior from dual perspectives, utilizing tools such as the Student–Teacher Relationship Scale and Conners’ Comprehensive Behavior Rating Scales-Teacher Assessment Report. Visual analysis and statistical analysis results indicate a strong positive impact of Banking Time on child–teacher relationships and a slight inhibitory effect on child problem behaviors. The implementation of Banking Time provides valuable insights into specific paths and strategies for promoting teachers’ professional development.

## 1. Introduction

Child problem behaviors refer to abnormal behaviors that hinder their adaptation to society [1]. Among the 1.5 to 6-year-old age group, 13% to 18% of children exhibit problem behaviors, and timely intervention in these children can help prevent potential lifelong psychological issues [2]. Furthermore, research unveiled that the COVID-19 pandemic caused significant changes in people’s lives, encompassing kindergarten closures, the adoption of online classes, a reduction in family income, and lifestyle adjustments, collectively contributed to an escalation in children’s problem behaviors [3,4,5,6]. The prevention and intervention of children’s problem behaviors have always been a focal point in various sectors of society, and it is especially important to improve problem behaviors in young children.

There are numerous methods to reduce child problem behaviors, among which, play therapy is widely applied, highly effective, and aligns with children’s natural inclination to enjoy games, making it suitable for all children [7]. The development of play therapy has progressed rapidly, initially involving interactions between doctors and children in specialized play therapy rooms. However, subsequent research indicated that if a person with an intimate relationship with the child replaces the doctor in the interaction, the therapeutic effect will be better due to the reliance on this close relationship to reduce problem behaviors [8]. As a result, it gradually evolved into interactions between parents or teachers, who receive training, and children at home or in kindergarten. Examples of such approaches include Teacher–Child Interaction Training (TCIT), Filial Therapy, Parent–child Interaction Therapy (PCIT), among others [9,10,11].

### 1.1. The Importance of Positive Child–Teacher Relationships

For children, child–teacher relationships have a significant impact on their long-term development [12]. The quality of child–teacher relationships influences problem behaviors in children, and such quality can predict the occurrence of externalizing problem behaviors [13]. Research indicates that an intimate child–teacher relationship can buffer the impact of adverse factors in a child’s development, while a dependent child–teacher relationship may exacerbate the impact of adverse factors during a child’s development [14]. Additionally, a conflicted child–teacher relationship may contribute to more severe problem behaviors in children [15]. Furthermore, conflicts in child–teacher relationships are more likely to occur between teachers and children exhibiting more externalizing problem behaviors [16]. Moreover, the quality of child–teacher relationships influences children’s academic success [17]. Child–teacher relationships have become a core factor influencing the improvement in educational quality [18].

For early childhood teachers, the quality of positive child–teacher relationships is particularly important. Teachers with low-quality child–teacher relationships are prone to emotional exhaustion [19]. On the other hand, teachers with good child–teacher relationships tend to experience high personal achievement [20].

Pianta, based on the developmental systems theory, proposed a conceptual model for child–teacher relationships, in which individual characteristics, interactive processes, and external influences all impact these relationships [21]. Among these factors, teacher characteristics are more amenable to change. Therefore, providing professional support to teachers can enhance the quality of child–teacher relationships [22].

### 1.2. Banking Time Intervention

Banking Time is an adaptation of Parent–Child Interaction Therapy, which falls under the category of play therapy [23]. Play therapy is widely used for intervening in problem behaviors in young children, showing significant effectiveness. It caters to children’s natural inclination toward play and is considered applicable to all children [7].

Banking Time, developed by Pianta, Hamre, and others, provides a series of child–teacher interaction methods to enhance the quality of the teacher–child relationship and address child problem behaviors [24]. According to attachment theory, the teacher’s sensitivity to and response to the child’s needs are crucial factors, even determinants, affecting the quality of the child–teacher relationship [25]. Banking Time improves teacher–child interaction by enhancing mutual understanding and interaction patterns between teachers and children. It emphasizes that the established intimate and supportive child–teacher relationship is an investment, serving as a resource to help teachers and children resolve conflicts and challenges during interactions. Therefore, it is referred to as Banking Time.

Banking Time has the following characteristics: Firstly, it involves one teacher and one child. Secondly, it takes place in a private, fixed, and disturbance-free location. Thirdly, the session is 10 to 15 min, with a frequency of 2 to 3 times per week, spanning 3 to 8 weeks as a major cycle. Fourthly, the content of sessions is freely chosen by the child, and it is entirely child-led. Fifthly, the teacher’s main objective is to actively engage, respond, and listen to the child during the child-initiated sessions, using the four methods of observing, narrating, labeling, and developing relational themes, as shown in Table 1. Observing involves carefully noting the behavior, speech, and feelings of the child, as well as the teacher’s own thoughts and feelings. Narrating encompasses three methods. The first is the “Sportscaster” technique, where the teacher loudly describes what the child is doing in an interested tone, akin to a sports commentator. For instance, if a child is building with blocks, the teacher might exclaim, “Wow, you connected a red block to the green one!” The second method, “Imitation”, entails nonverbal communication between the teacher and child, where the teacher mimics the child’s primary actions. For example, if a child uses a yellow crayon to draw a yellow house, the teacher sits beside the child and uses a green crayon to draw a green house. The third technique is “Reflection”, where the teacher partially modifies the child’s words. For instance, if a child proudly says, “Look! I made a lollipop with blocks”, the teacher responds with interest, “Wow! You made a big lollipop with pink and yellow blocks!” Labeling involves identifying the emotional state of the child. For example, if a child throws down a toy and says, “I can’t play with it”, the teacher labels this emotion as frustration and says, “Sweetie, you put the toy on the floor, you seem a bit frustrated”. Developing relationship themes entails teachers identifying words or sentences that can promote the quality of the child–teacher relationship after spending time with the child. These themes should align with the emotional experiences of both teachers and children during Banking Time sessions and throughout the day in kindergarten. They can include phrases such as “I’m interested in you”, “I’m always here”, “I respect you”, “You have the right to decide”, “I’m reliable”, “I can help you”, “You’re safe with me”, “I care about what you say”.

Additionally, teachers receive training from experts before implementing Banking Time sessions. Prior to implementation, parents and the child should be informed. In contrast to traditional child–teacher interactions, certain teacher behaviors are prohibited during Banking Time sessions, such as criticizing or praising the child and using the opportunity to teach knowledge or train specific skills.

The impact of Banking Time on child–teacher relationships and the improvement in children’s problem behaviors has been studied in specific groups. Driscoll and Pianta conducted a validation of the effectiveness of Banking Time on child–teacher relationships and the enhancement of children’s [26]. In the Head Start project, 29 teachers and 116 children were divided into a control group and an experimental group. Through the examination of STRS, it was found that teachers assigned to Banking Time scored higher in child–teacher relationship scores compared to the control group. Moreover, certain dimensions of problem behaviors, such as children’s frustration tolerance, task orientation, and problem behaviors, showed improvement. In this study, some aspects of children’s problem behaviors and the quality of child–teacher interactions were enhanced. However, because the participating teachers were also involved in rating children’s behavior, and no additional observers were set up for the collection of various data, there might be a subjective weakening or strengthening of the perception of real changes in children’s data during the scoring process.

Driscoll, Wang, and others conducted a study on the impact of teachers’ implementation of Banking Time on experimental outcomes [27]. The study involved 252 preschool teachers participating in a web-based professional development program for early childhood educators. In each classroom, approximately four children were randomly selected to assess the intervention’s effects. The study examined the influence of Banking Time on children’s language, literacy, and socioemotional development. The results showed that over the entire school year, children who participated in Banking Time had closer relationships with their teachers compared to those who did not participate. Participation in Banking Time also led to changes in children’s social behaviors, particularly in terms of social competence. Due to the study’s duration, language and literacy development in children could not be verified. However, similar to the previous study, this research encountered challenges during the scoring process. The same teachers were asked to submit scores for child–teacher relationships and children’s behavior at the beginning and end of the school year, potentially introducing scorer bias and inaccuracies in the experimental results.

Vancraeyveldt, Verschueren, and others combined Banking Time with Behavior Management Techniques to address and enhance preschoolers’ problem behaviors [28]. They observed a significant decrease in teachers’ ratings of externalizing behavior in 175 children. Further analyses revealed that the implementation of Banking Time had a positive impact on improving externalizing behavior in preschoolers.

Williford, LoCasale-Crouch, and others conducted a study with 470 preschoolers exhibiting high levels of externalizing behavior to test the impact of Banking Time [29]. All children were randomly assigned to one of three groups: the Banking Time group, the Child Time group (where there were no restrictions on child–teacher interactions), and a control group. In terms of teacher-reported child behavior, there was a reduction in disruptive behavior during both Banking Time and Child Time from baseline to the end of the experiment. As for observed teacher behavior during Banking Time, the experiment’s effectiveness was influenced if teachers exhibited lower negativity in their interactions with the children. In addition, the study measured cortisol levels, a key hormone aiding the body’s response to stress, from a physiological perspective. The researchers found a significant decrease in cortisol levels, indicating reduced stress response, only in children who experienced Banking Time intervention, with no changes observed in children under other conditions. However, the researchers noted that both Banking Time and Child Time had some positive effects on certain aspects of preschoolers’ behavior. Therefore, it cannot be conclusively proven that Banking Time necessarily has inhibitory effects on preschoolers’ problem behaviors, necessitating further research for a more comprehensive and accurate assessment of the impact of Banking Time on preschoolers’ behavior.

However, in some studies, the effectiveness of Banking Time is not as expected. Attwood conducted research with three groups of teachers from different racial backgrounds and 6- to 7-year-old children, using a multiple-baseline design across participants, indicating that Banking Time had no effect on children’s problem behaviors or child–teacher relationships [30]. Strand conducted a 2 × 2 mixed-factor experiment with 90 children and their teachers to investigate the impact of Banking Time and found that Banking Time had an effect on problem behaviors but not on child–teacher relationships [31]. Sahin conducted a single-factor between-subjects experiment with 93 children and 8 teachers to verify the impact of Banking Time on child–teacher relationships. The study revealed a significant impact of Banking Time on children’s perceived relationships but no significant impact on teachers’ perceived relationships [32]. In these three studies, researchers used multi-perspective measurement tools to examine the experimental results for mutual confirmation. However, limitations in experimental design, incomplete teacher cooperation, and external factors such as family dynamics may have influenced the experimental results and deviated from the hypotheses.

### 1.3. The Current Study

The purpose of the current study was to investigate its impact on the child–teacher relationship, child problem behaviors using an experimental design. However, evaluation results from different perspectives may vary [33]. Moreover, numerous studies indicate that both self-report and other-report, as well as observer-rated data, are indispensable in measurement [34,35,36]. Therefore, to make the research results more comprehensive and objective, this study adopts a dual-perspective approach, recording changes in child–teacher relationships from the perspectives of both children and teachers. Similarly, changes in child problem behaviors are recorded from the perspectives of both teachers and researchers.

The study hypothesized the following:

(a) Following the Banking Time intervention, both teachers and children perceive an improvement in the quality of teacher–child relationships.

(b) Following the Banking Time intervention, researchers and teachers perceive a decrease in problem behaviors.

## 2. Materials and Methods

This study employs a multiple-baseline experiment across subjects’ design, with Banking Time as the independent variable and child–teacher relationships, as well as child problem behaviors, as dependent variables. The aim is to investigate the impact of Banking Time on both child–teacher relationships and child problem behaviors.

### 2.1. Participants

This study employed nonrandom sampling. First, researchers extended invitations to the principals of four kindergartens in Guangdong Province of China, and all of them expressed a willingness to participate. Subsequently, through the recruitment of teachers, two teachers from each kindergarten were selected to participate in the study.

Then, teachers nominated three children based on the researcher’s descriptions, including characteristics such as “opposes you”, “does not initiate interaction with you”, and “cries or fusses”, and used the Student–Teacher Relationship Scale (STRS) for scoring. If the scores for all children were below 90 points, the child with the lowest score was chosen as a participant. If all scores were above 90 points, participation in the experiment was not recommended, and a different child or teacher needed to be selected.

In the study, a total of eight children and eight teachers were selected, with each child and their teacher forming a dyad. Detailed basic information can be found in Table 2.

### 2.2. Instruments

#### 2.2.1. Student–Teacher Relationship Scale

This study utilized STRS, compiled by Pianta, focusing on the intimacy and conflict subscales [37]. All descriptions were modified according to the relevant context and included an additional deception question, totaling 23 items. The scale used a Likert-style five-point scoring system, ranging from “definitely does not apply” to “definitely applies”. The reliability and validity of the scale are good in China [38]. The Cronbach’s alpha coefficient for this scale in the current study was 0.84, indicating good internal consistency. This scale is used to measure teachers’ perception of the child–teacher relationship, and teachers fill out the scale once a week.

#### 2.2.2. Conners’ Comprehensive Behavior Rating Scales-Teacher Assessment Report

This study employed Conners’ Comprehensive Behavior Rating Scales-Teacher Assessment Report (TAR). The TAR has been in use for over 50 years and is one of the most widely used questionnaires for screening problem behaviors in young children. All descriptions were modified according to the relevant context, resulting in a total of 28 items. It used a Likert-style four-point scoring system, ranging from “not true at all” to “very much true” [39]. This scale covers common problem behaviors in preschoolers and has been widely tested for reliability and validity in China [40]. The Cronbach’s alpha coefficient for this scale in the current study was 0.90, indicating good internal consistency. Teachers use this tool to assess their perceptions of child problem behaviors, and assistant teachers fill it out once a week.

#### 2.2.3. Child Behavior Observation Form

The Child Behavior Observation Form used in this study was adapted from the questions in the TAR and served as a complementary tool for validating the results. The researchers coded 30 min of behavior videos for each child using a time-sampling method. The videos included 10 min of group activities, 10 min of free activities, and 10 min of daily activities. If the duration of the target behavior was less than 10 s, it was recorded as 1 occurrence. If the duration exceeded 10 s, the frequency increased by 1 for each additional 10 s, rounded up to the nearest 10 s. This tool is used to measure researchers’ perceptions of child problem behaviors, and researchers fill out the form once a week.

#### 2.2.4. Child’s Perceived Child–Teacher Relationship Questionnaire

The Children Perceived Child–Teacher Relationship Questionnaire used in this study was adapted from Strand’s Child Questionnaire for children under 8 years old, combining it with the Chinese context [31]. The evaluation method involved placing two seemingly identical dolls with opposite meanings in front of the child. The researcher introduced the dolls, stating, “This one (one doll) is your friend, and this one (the other doll) is not your friend. Which one is Teacher Z?” Based on the child’s actual feelings, they selected a doll, and the researcher recorded the child’s answer. The questionnaire consisted of 11 items, and each selection by the child represented a positive answer (liking, happiness, helping, not afraid, not angry, praising, caring, assistance, playing, interesting, and is), scoring 1 point. This tool is used to measure children’s perceptions of the child–teacher relationship, and children completed the questionnaire before and after the implementation of Banking Time.

#### 2.2.5. Banking Time Social Validity Questionnaire

This study developed the Banking Time Social Validity Questionnaire to investigate teachers’ perceptions, opinions, and demographic information regarding Banking Time, comprising 13 items. Twelve questions are scored on a four-point Likert scale, ranging from “strongly agree” to “strongly disagree”. Teachers read the relevant descriptions and rate their level of agreement. One question is open-ended. Teachers fill out the questionnaire after completing the project.

#### 2.2.6. Experimental Fidelity Check Form

The Experimental Fidelity Check Form in this study was developed based on the implementation standards of Banking Time, consisting of 9 items, with 1 point awarded for each compliant item, totaling 9 points. It is used to record the implementation of Banking Time by evaluating videos submitted by teachers. The results are presented as a percentage.

#### 2.2.7. One-on-One Teacher–Child Interaction: Banking Time Teacher Manual

The One-on-One Teacher–Child Interaction: Banking Time Teacher Manual in this study was adapted from Pianta’s Banking Time Educator’s Manual [41]. It was edited in a popularized language format, covering sections such as “A Letter to Teachers”, “Introducing Banking Time to Parents”, “Introducing Banking Time to a Child”, “Planning Banking Time Sessions”, “Implementing Banking Time Sessions”, and “Recording Banking Time”. This manual served as a textual resource for teachers to implement and document Banking Time sessions.

### 2.3. Procedure

Following a multiple-baseline experiment design, Banking Time was implemented at different times in the three kindergartens to prevent changes in teachers’ perceptions of child–teacher relationships and childhood behavior after receiving training, which could affect data collection during the baseline period. The researcher conducted face-to-face training for teachers in the week before the implementation of Banking Time. The average training duration was about 60 min, covering the essence of Banking Time, implementation procedures, and simulated implementation. Dyad 7 and Dyad 8 from kindergarten 4 were designated as the control group and did not undergo any treatment.

Each teacher, along with their corresponding child, was considered a dyad. The start and end times of the Banking Time intervention were the same for dyads in the same kindergarten, with the start time randomly assigned for each kindergarten. The specific implementation period is shown in Figure 1. In addition, this study protected the privacy of participants, allowed them to withdraw at any time, and obtained informed consent from parents, along with verbal consent from the children.

### 2.4. Data Analysis

#### 2.4.1. Visual Analysis

Visual analysis is the most fundamental and intuitive method for analyzing data in a multiple baseline experimental design. In this study, all the data intended for analysis have been graphed to allow for subjective interpretation.

#### 2.4.2. Statistical Analysis

In addition to visual analysis, rigorous and statistically powerful methods are employed for objective validation.

For instance, a portion of the data collected and analyzed in a time series according to a multiple-baseline design includes scores such as teacher-perceived child–teacher relationship scores, teacher-perceived child behavior problem scores, and researcher-observed child behavior problem scores. These scores are analyzed using the Tau-U measure, proposed by Parker, which is suitable for single-case experimental designs (SCED) with baseline data showing trends or extreme values, as detailed in Formulas (1) and (2) [42]. The value of *S* is calculated through the Kendall rank correlation test. The Tau-U value ranges between 0 and 1. If it is less than 0.65, it indicates a weak effect. If it falls between 0.66 and 0.92, it suggests a moderate effect. If it exceeds 0.93, it signifies a strong effect [43].To facilitate scholars’ use, Pustejovsky and colleagues have developed a calculator specifically designed for Tau-U measure, which was employed in this study for analysis [44].
Tau-U = *S*/Pairs(1)
Pairs = *n*_A_ × *n*_B_(2)

Another portion of the data, collected and analyzed before and after the experiment, such as child-perceived child–teacher relationship scores, is subjected to paired-sample *t*-test analysis. The data in this study were analyzed using SPSS version 26.00.

### 2.5. Research Validity

The video observations in this study consisted of two parts: videos depicting teachers implementing Banking Time and videos capturing children’s behavior. After establishing observation criteria, one researcher randomly distributed 20% of the videos depicting teachers implementing Banking Time and children’s behavior to another researcher for coding. The consistency of coding between the two parts of videos was 86.11%, meeting the research requirements.

The implementation of Banking Time by teachers was recorded using the Experimental Fidelity Check Form filled out by researchers. The fidelity level of all teachers in implementing the intervention was greater than 70%, meeting the research requirements.

## 3. Results

### 3.1. Changes in Child–Teacher Relationships

#### 3.1.1. Changes in Teacher-Perceived Child–Teacher Relationships

In Figure 2, teachers engaged in the implementation of Banking Time perceived higher quality of child–teacher relationships during the experimental period compared to their baseline perceptions. Teachers who did not participate in Banking Time maintained a stable perception of the child–teacher relationship during the baseline period. Specifically, teachers in Dyads 1, 2, and 4 exhibited a noticeable increase in their perceived child–teacher relationship scores at different stages. Teachers in Dyads 3, 5, and 6 demonstrated a gradual upward trend in their perceived child–teacher relationship scores, while teachers in Dyads 7 and 8 maintained stable scores at different stages.

As depicted in Table 3, the overall Tau-U value stood at 0.86 (*p* < 0.001), signifying an exceptionally significant statistical difference. This implies that Banking Time has a notably positive impact on teachers’ perceptions of child–teacher relationships. Specifically, the Tau-U values for Dyads 1, 2, 3, 4, and 5 all reached 1.00, denoting a robust positive effect of Banking Time on the perceived child–teacher relationships for these five teachers. However, the Tau-U value for Dyad 6 was 0.33, indicating a relatively weak positive effect of Banking Time on the perceived child–teacher relationship for this particular teacher.

#### 3.1.2. Changes in Child-Perceived Child–Teacher Relationships

As shown in Figure 3, children who participated in Banking Time sessions perceived higher child–teacher relationships in the post-test compared to the pre-test. Children who did not participate in Banking Time sessions showed no change in their perceived child–teacher relationships between the pre-test and post-test.

As indicated in Table 4 and Table 5, there was a significant difference in children’s perceived child–teacher relationship between the pre-test (M = 7.83, SD = 2.04) and the post-test (M = 9.50, SD = 1.76). t (5) = −5.00, *p* = 0.00, and the mean difference between the pre-test and post-test was M = −1.67, SD = 0.82. As indicated in Table 6, the effect size, measured by Cohen’s d, was d = 0.82, indicating a large effect. The 95% confidence interval for the mean difference ranged from −3.49 to −0.56. Thus, Banking Time had a significant influence on children’s perceived child–teacher relationship.

#### 3.1.3. Summary

Whether relying on visual analysis or Tau-U analysis, the implementation of Banking Time led to an enhancement in the quality of child–teacher relationships as perceived by teachers. Furthermore, based on both visual analysis and paired sample *t*-test analysis, the quality of child–teacher relationships perceived by children improved after the implementation of Banking Time.

### 3.2. Changes in Child Problem Behavior

#### 3.2.1. The Changes in Teacher’s Perception of Child Problem Behavior

As illustrated in Figure 4, teachers engaged in the implementation of Banking Time reported changes in child problem behavior during the experimental period that were inconsistent with those perceived during the baseline period. Conversely, teachers who did not participate in Banking Time maintained stable scores in their perception of child problem behavior throughout the baseline period. Specifically, teachers in Dyads 1 and 3 exhibited an increasing trend in the perception of child problem behavior scores across different periods. In contrast, teachers in Dyads 2, 4, 5, and 6 demonstrated a decreasing trend in the perception of child problem behavior scores over various periods. Teachers in Dyads 7 and 8 perceived a relatively stable trend in child problem behavior across different periods.

As presented in Table 7, the comprehensive Tau-U value is −0.21 (*p* > 0.05), signifying a lack of statistical significance. This implies that Banking Time exerts a modest negative influence on teachers’ perception of child problem behavior.

#### 3.2.2. The Changes in the Researcher’s Perception of Child Problem Behavior

From the researcher’s perspective, the alterations in child problem behavior observed during the experimental period among participants in Banking Time implementation were incongruent with those observed during the baseline period. As depicted in Figure 5, children who were not involved in Banking Time displayed consistent scores in their problem behaviors throughout the baseline period. Specifically, children in Dyads 1 and 5 demonstrated an increasing trend in problematic behavior scores across different periods, while those in Dyads 2, 3, 4, and 6 exhibited a decreasing trend in problematic behavior scores during various periods. Children in Dyads 7 and 8 displayed a relatively stable trend in problem behavior changes across different periods.

As depicted in Table 8, the overall Tau-U value is −0.27 (*p* > 0.05), indicating a statistically significant yet weak negative impact of Banking Time on the researcher’s perception of child problem behavior. More specifically, Dyad 2 exhibits a Tau-U value of −1.00, signifying a robust negative impact of Banking Time on the researcher’s perception of child problem behavior. Dyad 4 shows a Tau-U value of −0.80, suggesting a moderate negative impact of Banking Time on the researcher’s perception of child problem behavior. Dyad 3 and Dyad 6 have Tau-U values of −0.17 and −0.36, respectively, indicating a weak negative impact of Banking Time on the researcher’s perception of child problem behavior. On the other hand, Dyad 1 and Dyad 5 display Tau-U values of 0.50 and 0.14, respectively, suggesting a weak positive impact of Banking Time on the researcher’s perception of child problem behavior.

#### 3.2.3. Summary

Whether through visual analysis or the Tau-U measure, the incorporation of Banking Time led to a marginal decrease in teachers’ perception of child problem behavior. Furthermore, whether relying on visual analysis or the Tau-U measure, the implementation of Banking Time led to a marginal decrease in the researcher’s perception of child problem behavior.

### 3.3. Social Validity

The social validity of this experiment was assessed using the Banking Time Social Validity Questionnaire. Following the implementation of the intervention, six teachers completed the questionnaires. The mean score for the Banking Time Social Validity Questionnaire was 37.67, with a standard deviation of 3.59. The maximum score recorded was 44, while the minimum score was 32. All scores surpassed the theoretical midpoint of 30.00, indicating an overall high level of social validity.

All teachers were in unanimous agreement that Banking Time plays a significant role in enhancing the quality of child–teacher relationships and improving the overall management of children’s daily activities. Some teachers provided specific observations, stating, ‘Ordinarily, children wait for the teacher to pour milk. However, during Banking Time sessions, they poured it themselves and even took the initiative to clean up spills with a cloth. It’s a departure from the usual scenario where they simply stand there without knowing what to do’. Additionally, others highlighted increased interaction and communication, with one teacher noting, ‘We engage in more conversation and frequently seek assistance from me’.

However, a minority of teachers expressed the view that the implementation of Banking Time did not necessarily result in a closer child–teacher relationship. One teacher mentioned, “The time spent alone with the child is no different from usual, and I didn’t feel a closer bond”. Despite this, the majority of teachers believed that Banking Time contributes to a reduction in problem behaviors in children and has a positive impact on their behavior. Some suggested that to observe significant changes, the duration of the session should be extended: “There is an impact, but it may take a bit longer to see results, especially for normal children”. A few teachers opined that alternative methods might have a more substantial impact on child problem behavior: “I think collaborating with parents to develop strategies would be more effective in changing children’s behavior”.

Many teachers acknowledged facing challenges during the implementation of Banking Time but expressed satisfaction with the support they received. For instance, one teacher shared, “Initially, I was anxious. I told her we were going to a secret base, just the two of us. But she kept looking at me without responding until the fourth day when she finally interacted with me”. Another teacher mentioned, “At the beginning, the materials were insufficient, and the children weren’t very interested. After making adjustments, it became evident that the children became more engaged”.

Despite these challenges, all teachers believed they could independently carry out Banking Time as required. However, a few teachers raised concerns about the significant time commitment and its impact on their daily responsibilities: “During the session, the children were running around and uncooperative. Another teacher had to record the video, leaving me alone to take care of over twenty other children. It was challenging”.

## 4. Discussion

In this study, the effectiveness of Banking Time in enhancing the quality of child–teacher relationships has been successfully demonstrated through a multiple baseline experimental design, while the inhibitory effect of Banking Time on child problem behaviors was not confirmed successfully.

The findings of this study are consistent with previous studies in terms of teacher–child relationships [26,27]. The findings of this study are consistent with Attwood in terms of child problem behaviors [30]. According to play therapy, children with problem behaviors have difficulty getting along, and making them feel safe and cared for can help them improve their problem behaviors [45]. Banking Time provides a range of methods to help children feel safe and cared for, which may contribute to an improvement in children problem behaviors. However, when considering whether to apply Banking Time, it is necessary to take into account the limitations of this method as well as the individual differences of the participants in this study.

### 4.1. Limitation of Banking Time

Variability in implementation is common and unavoidable in educational settings [46]. The implementation of Banking Time is constrained by different physical environments. Firstly, Banking Time was developed in the United States and applied in countries such as the United States and Turkey, requiring teachers to implement it 2–3 times a week for 10–15 min each time. However, the teacher–child ratio and workload in Chinese kindergartens are different, making it challenging for teachers to plan and implement Banking Time sessions more frequently. Teacher 4 mentioned, ‘It’s too difficult to squeeze out time every day for this intervention. It might be more suitable for implementation in some one-on-one training institutions’. Secondly, the ideal condition for Banking Time sessions is a quiet and fixed location, which is not easy to achieve in Chinese kindergartens. Teacher 6 mentioned, ‘Today there is a loud rehearsal for the sports day outside, and Child 6 always runs over to watch.’

Unforeseeable factors have significantly influenced the experimental period, resulting in the absence of noticeable effects of Banking Time. Various uncontrollable events, such as “a child being unwell and unable to attend kindergarten”, “a child relocating to their hometown”, “a child experiencing isolation in the classroom this Wednesday”, and “the kindergarten undergoing evaluation this week with associated material preparation, causing a slight increase in workload”, have all disrupted the anticipated cycle of Banking Time. While these occurrences reflect real-life situations in the field, it is imperative to recognize that these uncontrollable factors have left an enduring impact on the efficacy of Banking Time. In real-world settings, the implementation of experiments is influenced by at least 23 contextual factors, and devising an effective intervention marks only the initial phase of success, as the subsequent process of implementing this intervention is complex and unfolds over the long term [47].

### 4.2. Differences at the Participant Level

External factors have a more significant negative impact on children’s problem behavior than the positive influence of Banking Time. Although research has demonstrated that child–teacher relationships can improve children’s problem behavior, there are numerous influencing factors. Apart from Banking Time, home-rearing, family instability, child temperament, child prosocial behavior, child language ability, and executive function have varying degrees of influence on children’s problem behavior [48,49,50,51,52]. Notably, factors originating from the family play a distinct and predictive role in the occurrence and persistence of child problem behavior [53]. Examples include: “Child 1’s parents are very strict with him, telling him they won’t pick him up if he doesn’t behave; they strongly endorse this parenting approach”. “Child 3, when young, lacked parental companionship and has now developed autism”. “Child 5 has a sister with autism who cannot be separated from her in kindergarten. The sister cannot tolerate unfamiliar environments and refuses rehabilitation training. When the sister is present, the child only stays with her, exhibiting unstable emotions. However, if the sister is absent, the child interacts with other children, engages in preferred activities, and experiences more stable emotions”. “Child 6 has a newborn sister, and his parents pay less attention to him”. In real-world scenarios, these influencing factors are challenging to eliminate; they are also incalculable. On the other hand, differences at the teacher level also exist. Research indicates a positive correlation between teachers’ level of engagement and improvements in teacher–child relationships, with teachers holding child-centered beliefs demonstrating higher quality interactions with children [54]. However, as this study did not involve measurements of teacher engagement levels and beliefs, this remains a potential possibility.

The variability in the data significantly influences the scoring outcomes. The impact of Banking Time on the perceived teacher–child relationship and child problem behavior hinges on the data provided by teachers and children. Divergent perceptions of the same issue at the same level in the scores may arise among different teachers and children. Various factors, including teachers’ educational background, work pressure, comprehension of the questionnaire’s inquiries, and children’s individual traits, expressive capabilities, and cognitive levels, can all contribute to shaping the research results. Notably, Mashburn, and Hamre found that 15% to 33% of scoring differences result from variations between scorers, with teachers’ evaluations of positive relationships and behaviors showing a positive correlation with high self-efficacy and a low teacher-student ratio, while also being influenced by factors such as race, classroom ambiance, and work environment [55].

### 4.3. Implication

This study holds significant importance, both in terms of its research content and methodology.

First and foremost, it contributes vital evidence regarding the impact of the Banking Time intervention. Going beyond previous research on Banking Time, this study delves into comprehensive investigations, showcasing the intervention’s effective enhancement of child–teacher relationships and its potential to address child problem behavior. Additionally, it identifies the implementation limitations of Banking Time, thereby enriching research outcomes, supplementing relevant literature, and providing valuable insights for the implementation and future research of Banking Time.

Secondly, this study contributes to the literature on child–teacher relationships, particularly from the children’s perspective. Children, often influenced by cognitive and emotional factors, typically find themselves in a less favorable position when it comes to evaluating child–teacher relationships. Research focusing on this dual-subject relationship, where both teachers and children assess each other’s relationships, is not commonly explored in previous studies. Furthermore, existing research highlights a notable inconsistency between teachers’ perceptions of child–teacher relationships and those perceived by children [56]. This study acknowledged the rights of both children and teachers to evaluate their relationships, utilizing scientific assessment tools to accurately measure the genuine impact of Banking Time. It also serves as a valuable reference for assessing child–teacher relationships from the children’s perspective.

Finally, this study contributes to the existing research on the application of the Tau-U measure. In SCED, Tau-U is considered one of the most popular and widely used indices with promising results [57,58,59]. This study, by combining the Tau-U measure with visual analysis and parameter testing, not only ensures the reliability of its research findings but also promotes the broader application of the Tau-U measure, enhancing the practical value of SCED.

### 4.4. Limitations of This Study

On one hand, there is a lack of consistency testing for teachers’ perceptions of child problem behavior. In this study, the impact of Banking Time on child behavior was examined from both the teacher’s and the researcher’s perspectives to verify whether the intervention influences child problem behavior. In each dyad, an additional teacher who did not implement Banking Time sessions was selected to assess the target child’s problem behavior, mitigating rater bias effects. To ensure rigor in the study, 20% of the child behavior videos were assessed by another researcher to calculate inter-rater reliability with the researcher, avoiding researcher expectancy effects. However, it was discovered during the actual data collection and processing that relying entirely on data from one teacher’s perspective on child problem behavior could be challenging to interpret if extreme values affecting the experimental results appeared in the ratings. Additionally, research suggests that caregivers of children with problematic behaviors are likely to overlook the child’s prosocial behaviors, emphasizing the problematic ones [60]. Therefore, ensuring the validity of teachers’ assessments of children’s behavior is particularly important. Including an additional teacher, even though two teachers might produce different results in rating the problematic behavior of the same child using the same tool, would, to some extent, corroborate whether the recorded changes in child behavior align with the actual situation, enhancing the persuasiveness of the research results.

On the other hand, uncontrollable factors affect the experimental period. Various uncontrollable factors, such as unexpected events like pandemics, children falling ill, child turnover, teacher illness, teacher exam preparation, and teacher turnover, can impact the experimental period. If the experimental period is affected, the implementation of Banking Time is influenced, and the collection of baseline data is affected, thereby impacting the analysis of the effects and analysis of Banking Time on child–teacher relationships and children’s problem behavior. Despite the clear definition of the experimental period during participant recruitment, with the expectation that teachers would choose children who are less likely to drop out, it is still impossible to avoid a shortened experimental period due to factors such as children falling ill. In this study, data scarcity in some dyads significantly affected Tau-U values, and *p*-values, influencing statistical significance and resulting in a deviation between teachers’ actual perceptions of children’s behavior and changes in child–teacher relationships and the submitted data. If it were possible to extend the experimental period and sample the dependent variables as much as possible, the analysis would be more scientifically sound.

### 4.5. Future Directions

Like other SCEDs, the generalization of findings is easily limited by the small sample size of the study [61]. For future research, to improve the external validity of Banking Time, it is necessary to replicate the experiment in a more diverse and larger population of children to investigate the effects of Banking Time in different scenarios and with different groups of children. Furthermore, while Banking Time enhances child–teacher relationships through short and efficient one-on-one interactions, it may be challenging to implement for every child in the same classroom. Moreover, the determining factors influencing the effectiveness of Banking Time still need exploration [62]. It remains important to explore the extent to which Banking Time can support teachers and change teacher and child behavior [63]. Future research could seek more flexible ways to discover and implement the core elements of Banking Time.

Moreover, the findings of this study suggest that the impact of Banking Time on child problem behavior is nuanced and varies among individuals. While the overall effect size is small, there are individual differences in how children respond to this intervention. Teachers should consider the unique characteristics and needs of each child when implementing Banking Time. Flexibility and adaptability in the application of this approach are crucial to maximizing its benefits. In terms of future research, it is recommended to explore the long-term effects of Banking Time. Longitudinal studies that track the progress of teacher-student relationships and children’s behaviors over an extended period can provide valuable insights into the sustained impact of this intervention. Individuals require some time to undergo intervention, and the effects generated by the intervention may also take some time to manifest [64]. Additionally, investigating the moderating factors that influence the effectiveness of Banking Time, such as child temperament, teacher characteristics, and the classroom environment, can contribute to a more comprehensive understanding of its applicability and limitations.

Finally, expanding the scope of research to include diverse cultural contexts and educational settings can enhance the generalizability of findings. The current study was conducted in a specific cultural and educational context, and future research should explore how Banking Time functions in different cultural and linguistic environments. Individuals’ cultural background, race, worldview, and communication style can impact the effectiveness of interventions [65]. This cross-cultural perspective can contribute to the development of culturally sensitive and globally applicable strategies for promoting positive child–teacher relationships.

## 5. Conclusions

This study employed a multiple baseline experimental design to explore the impact of Banking Time on child–teacher relationships and child problem behaviors. The results of the study indicate that Banking Time had a positive influence on child–teacher relationships, while its impact on child problem behavior was relatively limited. 

Overall, Banking Time exhibited the potential to enhance child–teacher relationships in this study. However, the subtle effects on child problem behaviors indicate the need for further exploration. By continually investigating and refining the application of Banking Time in diverse educational settings, educators can leverage its advantages to create supportive and positive learning environments for children.

## Figures and Tables

**Figure 1 behavsci-14-00213-f001:**
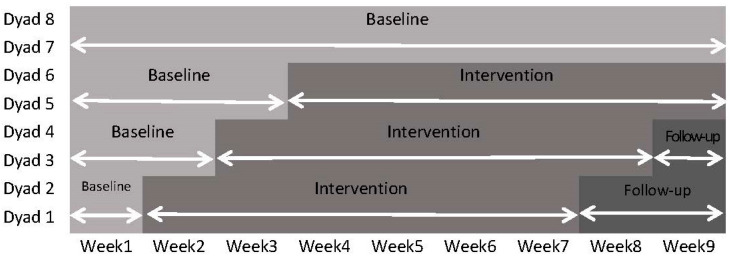
Implementation period.

**Figure 2 behavsci-14-00213-f002:**
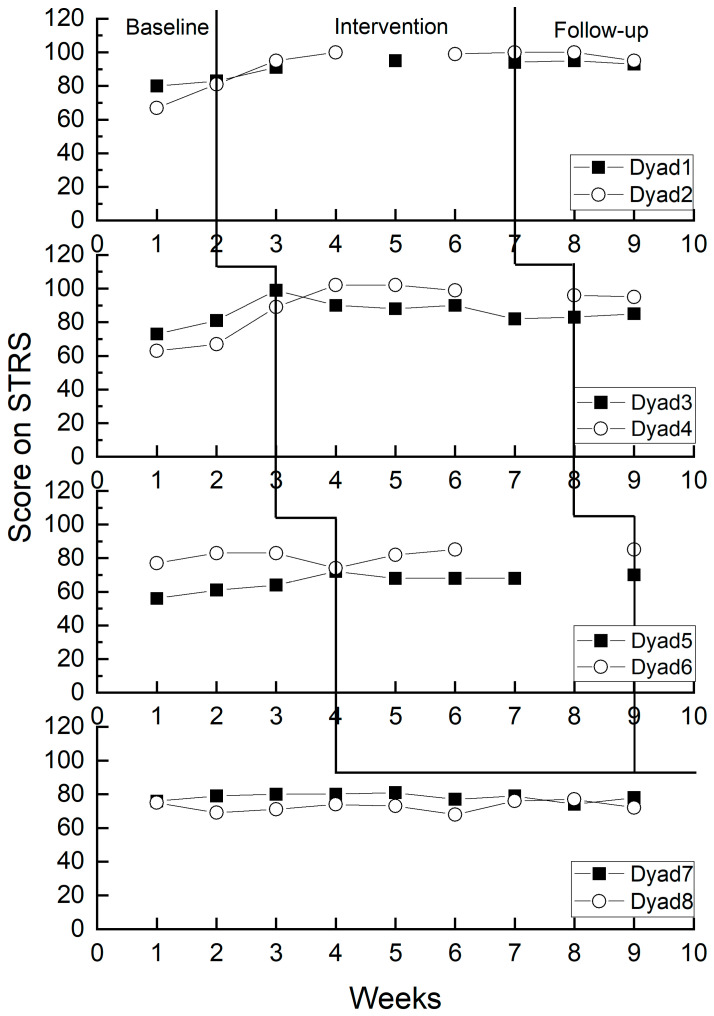
Changes in teacher-perceived child–teacher relationship.

**Figure 3 behavsci-14-00213-f003:**
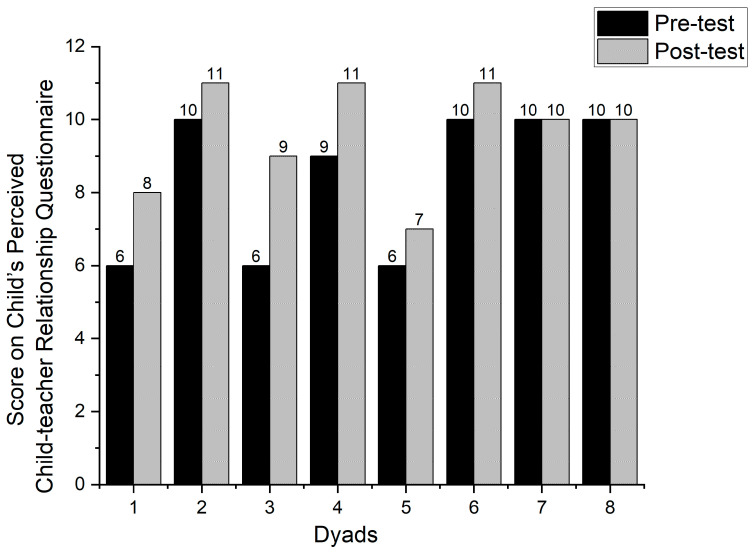
The changes in child–teacher relationship perceived by children.

**Figure 4 behavsci-14-00213-f004:**
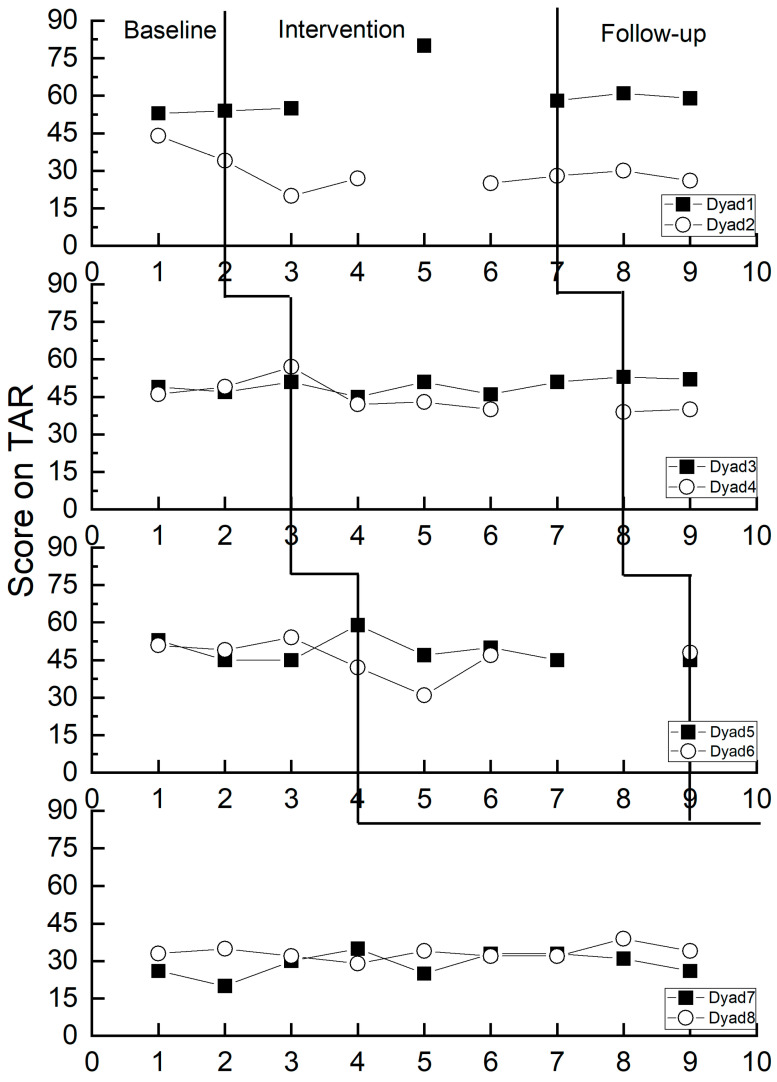
Changes in teacher-perceived child problem behavior.

**Figure 5 behavsci-14-00213-f005:**
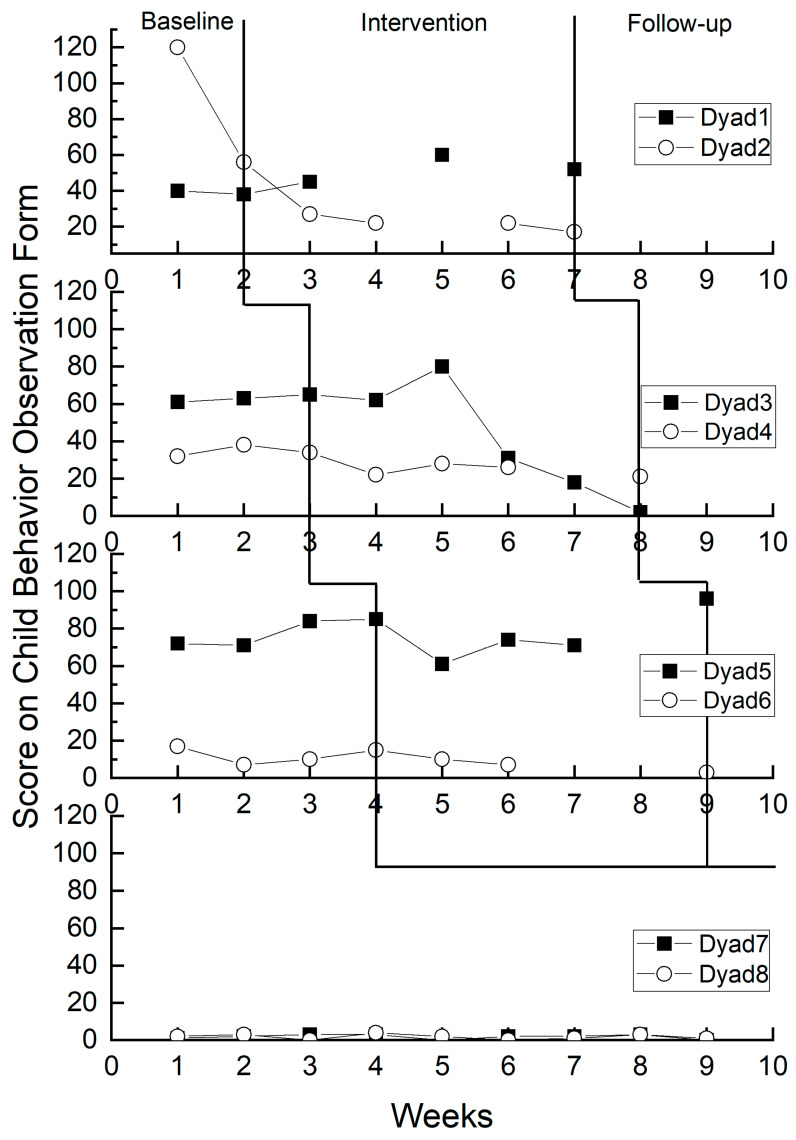
Changes in the researcher-perceived child problem behavior.

**Table 1 behavsci-14-00213-t001:** Banking Time techniques.

Techniques	Right	Wrong
Observing	Continuous attention	Distracted
Narrating	Sportscasting (Child doing—Teacher describing) Reflection (Child saying—Teacher saying) Imitation (Child doing—Teacher doing)	Asking Giving suggestions Imparting knowledge Instructing the child to follow game rules
Labeling	Objectively describe all of the child’s emotions	Asking about the child’s feelings
Developing Relational Themes	Choose a fixed relationship theme to develop	Stop developing relationship themes after Banking Time

**Table 2 behavsci-14-00213-t002:** Participants’ basic information.

Kindergarten	Dyad	Participant
1	1	Teacher 1, female, 37 years old, main teacher, 5 years of teaching experience.
		Child 1, male, 4 years old, STRS score 80.
	2	Teacher 2, female, 30 years old, main teacher, 9 years of teaching experience.
		Child 2, female, 4 years old, STRS score 67.
2	3	Teacher 3, female, 28 years old, main teacher, 4 years of teaching experience.
		Child 3, male, 4 years old, diagnosed with autism, STRS score 73.
	4	Teacher 4, female, 20 years old, assistant teacher, 2 months of teaching experience.
		Child 4, male, 4 years old, STRS score 63.
3	5	Teacher 5, female, 28 years old, main teacher, 6 years of teaching experience.
		Child 5, female, 4 years old, diagnosed with autism, STRS score 56.
	6	Teacher 6, female, 26 years old, main teacher, 5 years of teaching experience.
		Child 6, male, 4 years old, STRS score 77.
4	7	Teacher 7, female, 25 years old, main teacher, 5 years of teaching experience.
		Child 7, female, 4 years old, STRS score 86.
	8	Teacher 8, female, 24 years old, main teacher, 3 years of teaching experience.
		Child 8, female, 4 years old, STRS score 75.

**Table 3 behavsci-14-00213-t003:** Tau-U analysis of teacher-perceived child–teacher relationships for Banking Time.

Participant	*S*	Pairs	Tau-U	z	*p*
Dyad 1	4	4	1.00	1.41	0.15
Dyad 2	5	5	1.00	1.46	0.14
Dyad 3	12	12	1.00	2.00	0.05
Dyad 4	10	10	1.00	1.94	0.05
Dyad 5	15	15	1.00	2.24	0.03
Dyad 6	4	12	0.33	0.71	0.48
Total	50	58	0.86	3.98	0.00

**Table 4 behavsci-14-00213-t004:** Descriptive statistics of children’s perceived child–teacher relationship.

	Mean	N	StDev	SE Mean
Pre-test	7.83	6	2.04	0.83
Post-test	9.50	6	1.76	0.72

**Table 5 behavsci-14-00213-t005:** Paired samples test of children’s perceived child–teacher relationship.

Mean	StDev	SE Mean	t	df	*p*	Mean
Pre-test–Post-test	−1.67	0.82	0.33	−5.00	5.00	0.00

**Table 6 behavsci-14-00213-t006:** Paired samples effect sizes of children’s perceived child–teacher relationship.

		Standardizer	Point Estimate	95% Confidence Interval	
				Lower	Upper
Pre-test–Post-test	Cohen’s d	0.82	−2.04	−3.49	−0.56

**Table 7 behavsci-14-00213-t007:** Tau-U analysis of teacher-perceived child behavior problem by Banking Time.

	*S*	Pairs	Tau-U	z	*p*
Dyad 1	4	4	1.00	1.41	0.16
Dyad 2	−5	5	−1.00	−1.46	0.14
Dyad 3	4	12	0.33	0.67	0.51
Dyad 4	−6	10	−0.60	−1.16	0.25
Dyad 5	3	15	0.23	0.45	0.66
Dyad 6	−12	12	−1.00	−2.12	0.03
Total	−12	58	−0.21	−0.99	0.32

**Table 8 behavsci-14-00213-t008:** Tau-U analysis of Researcher-Perceived Child Behavior by Banking Time.

	*S*	Pairs	Tau-U	z	*p*
Dyad 1	2	4	0.50	0.71	0.48
Dyad 2	−5	5	−1.00	−1.46	0.14
Dyad 3	−2	12	−0.17	−0.33	0.74
Dyad 4	−8	10	−0.80	−1.55	0.12
Dyad 5	2	15	0.14	0.30	0.77
Dyad 6	−4	12	−0.36	−0.71	0.48
Total	−15	58	−0.27	−1.22	0.22

## Data Availability

The data presented in this study are available on reasonable request from the corresponding author.

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
