# Peer review of "Effect of Banking Time Intervention on Child–Teacher Relationships and Problem Behaviors in China: A Multiple Baseline Design"

_behavsci, 2024, doi:10.3390/bs14030213_

Round 1

Reviewer 1 Report (Previous Reviewer 3)

Comments and Suggestions for Authors

I thank the authors for incorporating my notes. Now gradually to those where I still have some reservations.

1. List of literature. Here, the authors really managed to incorporate more up-to-date sources, but the PRISMA model is still missing from the text.

2. Structuring is now fine.

3. Hypotheses. A hypothesis talks about the relationship of two or more variables on a general level. I therefore recommend explicitly mentioning the hypotheses on a separate line and not as part of the text.

4. Comments 6: Working with a t-test on a sample of 6 respondents is not suitable. This sample is very small and needs to be supplemented with an effect size with confidence intervals (SPSS allows this since version 27) Response 6: We acknowledge the identified issues and sincerely appreciate the suggested solutions. The revisions have been made in accordance with your recommendations, as outlined on page 10. In this case, we still do not understand each other. The authors provided a confidence interval, but I couldn't find anywhere Cohen's d representing substantive significance with interpretation.

Other notes have been incorporated. Now briefly what is required: i) PRISMA model, ii) explicit definition of hypotheses, iii) concrete measures of material significance with interpretation.

Author Response

Reviewer 2 Report (New Reviewer)

Comments and Suggestions for Authors

We value the courage of the research for the use of an experimental design and an observational methodology. The systematization with which they were applied is a merit that should be taken into consideration. In addition, the subject of the study has theoretical and applied relevance since its results will help to implement the Banking Time intervention more effectively.

However, we consider that the work presents a number of critical issues that should be improved. Firstly, in relation to the theoretical introduction, we miss a more general framing of the subject matter of the study within the line of research on teacher-student interaction. In the section on instruments, one of the most relevant authors, Robert C. Pianta, is cited in this line of work. Thus, we consider that it would be necessary to include other theoretical references on teacher-student interaction, which would help to contextualize the intervention methodology applied in the study.

We suggest taking into consideration a particularly interesting construct in the study of the teacher-student relationship, such as teacher emotional engagement, since it could offer insight into the processes underlying Banking Time.

In addition, we consider that there are insufficient theoretical references on the subject of behavioral problems in early childhood. Given that it is one of the variables under study, a broader theoretical vision of this problem should be available.

Regarding the theoretical references on the Banking Time methodology, we ask the authors for some work detailing the competencies involved in the dyadic interactions between the teacher and the student, so that they can be taken into account when discussing the results obtained in the observational study.

As for the method section, we consider it inadequate to explore the cultural adaptability of Banking Time and seek to verify the social validity of this methodology in China. With such a small sample, such a claim is completely unattainable, so we recommend that another more specific purpose of the study itself be established. For this reason, we also suggest that it not be included in the instruments section or in the results.

Regarding the specification of the variables, the quality of the teacher-student relationship should be included as a dependent variable, since it is a variable that is taken into account in the results and in fact is the one that shows significant improvements in the study.

Finally, in the discussion, sections 4.1 and 4.2 do not offer a synthesis of the results or a reflection on them, but rather introduce a series of variables that have not been studied in the research. We consider that this section should establish a more explicit relationship between the introductory theoretical framework and the results obtained.

Round 2

Reviewer 1 Report (Previous Reviewer 3)

Comments and Suggestions for Authors

Thanks for the authors' responses. I agree that the PRISMA model is often used in review articles. However, it is also often used when it is necessary to have an overview of the given issue, especially in relation to theory. I drew attention to the given model for the reason that the selection of literature seems random. However, I accept the authors' argument. I consider the incorporation of the other comments to be sufficient.

Reviewer 2 Report (New Reviewer)

Comments and Suggestions for Authors

The improvements introduced by the authors, based on the indications made by this reviewer, improve the article; we therefore recommend its publication.

This manuscript is a resubmission of an earlier submission. The following is a list of the peer review reports and author responses from that submission.

Round 1

Reviewer 1 Report

Comments and Suggestions for Authors

- The abstract should identify the study tools and additional data analysis.

- The abstract may specify/expand further on these eight children's levels.

- In the experimental process, tables 2 and 3 should be crossed together to show which teachers take care of which students. This will make the experimental process more detailed and clear.

- Acquisition of participants in this research: It appears that the method of acquisition has not yet been specified. (random/non-random) of such groups Does the researcher know that obtaining such groups affects the reliability of research results obtained from using reference statistics (paired sample t-test)?

- The absence of participating groups in Table 5 should be explained or further clarified. This may be mentioned in a discussion of the results or limitations of the research.

- Summary of research results: The results should be summarized by categorizing them according to the research objectives/hypotheses. This will make it easier and faster for readers to understand.

Reviewer 2 Report

Comments and Suggestions for Authors

lines 23 to 28. The consistency in relation to the pedagogical effects of COVID can be reinforced by the research of https://scholar.google.com/scholar_lookup?title=La+mediaci%C3%B3n+did%C3%A1ctica+del+profesorado+italiano+de+ELE+ante+el+desaf%C3%ADo+del+confinamiento+domiciliario+por+COVID-19&author=Ortega,+J.H.&author=Collado,+J.R.&publication_year=2022&pages=1383%E2%80%931411

Lines 59 to 72. Perhaps if a list of the problems that are announced were included (since different studies are referred to) a more descriptive perspective of the announced situation could be obtained.

Lines 79 to 81. Academic studies should be provided to contextualise this methodology.

The graphics on page 7 (after line 279) need to be edited. As there is not enough space between the words WEEK1WEEK2.... makes them difficult to read. Consider whether it is appropriate to change the spacing between them, or replace them with the formula W1 for WEEK1, W2 for WEEK2, etc.

Page 8, after line 303, make the same change as for the graphics on the previous page. D1 for DAY1, D2 for DAY2...

Repeat editing with graphics on page 9, 10, 

Reviewer 3 Report

Comments and Suggestions for Authors

Thank you for the opportunity to become a reviewer of the submitted article. As was the case in previous reviews, I will write my observations in bullet points for greater clarity to the authors:

1. Work with literary sources.

A large number of literary sources are out of date. It is evident that these are randomly chosen literary sources. In order for the theoretical background to be sufficient, it is necessary to build a prism model, insert it into the text and then proceed from it.

2. The theoretical part is not sufficiently structured. Basically, this is a long chapter. I recommend breaking down the text more so that it is clear from which sources it is based and why it is important to discuss the data.

3. Research hypotheses are not well formulated.

4. It is not clear how the respondents were chosen. Was it random sampling, as is necessary in hypothesis testing?

5. The research sample was too small to allow hypothesis testing and generalization of conclusions

6. Working with a t-test on a sample of 6 respondents is not suitable. This sample is very small and needs to be supplemented with an effect size with confidence intervals (SPSS allows this since version 27)

7. The above-mentioned shortcomings are not discussed at all within the limits of the research.
